# Detection of *XPO1*^E571K^ Gene Mutation from Cell-Free DNA in Blood Circulation of Lymphoma Patients by FAST-COLD PCR

**DOI:** 10.3390/ijms26157324

**Published:** 2025-07-29

**Authors:** Suwit Duangmano, Natsima Viriyaadhammaa, Pinyaphat Khamphikham, Nutjeera Intasai, Adisak Tantiworawit, Teerada Daroontum, Sawitree Chiampanichayakul, Songyot Anuchapreeda

**Affiliations:** 1Department of Medical Technology, Faculty of Associated Medical Sciences, Chiang Mai University, Chiang Mai 50200, Thailand; suwit.du@cmu.ac.th (S.D.); natsima.v@cmu.ac.th (N.V.); pinyaphat.kha@cmu.ac.th (P.K.); nutjeera.in@cmu.ac.th (N.I.); 2Cancer Research Unit of Associated Medical Sciences (AMS CRU), Faculty of Associated Medical Sciences, Chiang Mai University, Chiang Mai 50200, Thailand; 3Department of Internal Medicine, Faculty of Medicine, Chiang Mai University, Chiang Mai 50200, Thailand; atantiwo@yahoo.com; 4Department of Pathology, Faculty of Medicine, Chiang Mai University, Chiang Mai 50200, Thailand; teerada.k@cmu.ac.th; 5Center of Excellence in Pharmaceutical Nanotechnology, Chiang Mai University, Chiang Mai 50200, Thailand

**Keywords:** cell-free DNA, FAST-COLD-PCR, cancer, lymphoma, *XPO1^E571K^*

## Abstract

The *XPO1* (*exportin 1*) gene encodes exportin 1 protein responsible for transporting proteins and RNA from the nucleus to the cytoplasm. It has been used as a biomarker for lymphoma detection. *XPO1*^E571K^ mutation has been frequently observed and identified as a good prognostic indicator for lymphoma patients. The detection of a target molecule released by lymphoma cells into blood circulation (cell-free circulating tumor DNA, cfDNA) is a better method than tissue biopsy. However, cfDNA concentration in blood circulation is very low in cancer patients. Therefore, a precise and sensitive method is needed. In this study, cfDNA was extracted, and then the *XPO1* gene was detected and amplified using conventional PCR. Sanger sequencing was employed to verify the DNA sequences. FAST-COLD-PCR was developed to detect *XPO1*^E571K^ gene mutation using a CFX96 Touch Real-Time PCR System. The optimal critical temperature (Tc) was 73.3 °C, allowing selective amplification of *XPO1*^E571K^ mutant DNA while wild-type *XPO1* could not be amplified. *XPO1*^E571K^ gene mutation can be detected by this method with high specificity and sensitivity in lymphoma patients. This approach facilitates rapid and straightforward detection in a timely manner after the diagnosis. Accordingly, the optimized FAST-COLD-PCR conditions can be used as a prototype for *XPO1*^E571K^ mutant detection in lymphoma patients.

## 1. Introduction

Lymphoma is a heterogenous group of malignancies arising from lymphatic tissues and accounts for approximately 3–4% of all cancer cases worldwide. The incidence of non-Hodgkin lymphoma (NHL) has been increasing with diffuse large B-cell lymphoma (DLBCL, NOS) being the most common type. In Thailand, 4056 new cases of NHL were reported between 2007 and 2014, with DLBCL accounting for 58.1% [1,2]. Most lymphomas (85–95%) are derived from B lymphocytes, which are responsible for antibody production and form a key part of the adaptive immune response [3]. The pathogenesis of aggressive B-cell lymphomas such as Burkitt’s lymphomas (BLs) involves the dysregulation of the *c-Myc* proto-oncogene. Located on chromosome 8q24, c-Myc regulates diverse cellular processes including proliferation and apoptosis. In BL, c-Myc is translocated to immunoglobulin loci, resulting in its aberrant overexpression and uncontrolled B-cell proliferation [4,5].

Exportin 1 (XPO1), also known as Chromosome Region Maintenance 1 (CRM1), is a nuclear export protein that mediates the translocation of numerous molecules including tumor suppressors such as p53, BRCA1, Survivin, NPM, APC, and FOXO [6]. Overexpression of mutation of XPO1 has been implicated in various hematologic malignancies including chronic lymphocytic leukemia (CLL), primary mediastinal B-cell lymphoma (PMBL), classical Hodgkin lymphoma (cHL), and DLBCL [6,7,8]. A recurrent point mutation, *XPO1*^E571K^, involving a glutamate-to-lysine substitution at codon 571 (chr2:g.61718472C>T), is particularly common in PMBL (28%) and HL (26%) [9]. This mutation is believed to alter the binding affinity of XPO1 with nuclear cargo proteins and contribute to oncogenic transformation. The strategic role of XPO1 in cellular homeostasis and the recurrent nature of this specific mutation make *XPO1*^E571K^ an attractive target for both therapeutic intervention and diagnostic applications.

Traditionally, lymphoma diagnosis relies on clinical evaluation, lymph node biopsy, and various blood tests including CBC, LDH, and ESR. Bone marrow aspiration is reserved for advanced disease. In recent years, the analysis of cfDNA from peripheral blood has emerged as promising non-invasive method for cancer diagnosis and monitoring [10,11]. CfDNA originates from apoptotic or necrotic cells and carries genetic information representative of tumor burden. It has been utilized for early cancer detection, prognosis, and assessment of treatment response [12,13,14].

Several methods have been developed to discriminate between wild-type and mutant alleles of specific genes, particularly in liquid biopsy contexts. Sanger sequencing is still widely used for mutation detection, although its sensitivity is limited (~15–20% mutant allele frequency). More sensitive approaches such as allele-specific PCR (AS-PCR), real-time PCR with TaqMan probes, and droplet digital PCR (ddPCR) offer improved detection of low abundance mutations, including those in cfDNA [15,16,17]. While ddPCR can detect mutations at frequencies as low as 0.1%, the technique requires expensive instrumentation and is limited to analyzing a small number of targets per reaction [18,19]. For the *XPO1*^E571K^ mutation specifically, digital PCR (dPCR) and next-generation sequencing (NGS) have been applied in both research and clinical settings [9]. However, these techniques are often cost-prohibitive and require sophisticated equipment. Allele-specific PCR (AS-PCR) methods utilize primers designed to preferentially amplify mutant sequences, offering cost-effective mutation detection. However, traditional AS-PCR approaches often lack the sensitivity required for cfDNA analysis and may suffer from non-specific amplification of wild-type sequences [20]. ARMS-PCR (amplification refractory mutation system) and COLD-PCR (co-amplification at lower denaturation temperature) techniques have been developed to improve mutation detection sensitivity. COLD-PCR, in particular, can enrich for mutation-containing sequences by exploiting differences in DNA melting temperatures, achieving sensitivity of 0.1–1% in various applications [21,22].

Fast co-amplification at lower denaturation temperature PCR (FAST-COLD-PCR) is a sensitive and cost-effective technique that enriches mutant alleles by exploiting the differential melting temperature between mutant and wild-type DNA [23]. This method is particularly useful in detecting rare somatic mutations in cfDNA, making it an attractive option for non-invasive mutation profiling in cancer patients.

So far, the early diagnosis of cancer remains a significant challenge for scientists. In the early stages, solid tumors were difficult to detect due to minimal biochemical and anatomical differences from normal cells. As a result, diagnostic tests may yield uncertain outcomes, including false negatives, overdiagnosis, and overtreatment [24,25]. New cfDNA diagnostic technology can help overcome the limitations due to its high sensitivity and specificity [26]. This study aimed to establish a FAST-COLD-PCR methodology specifically designed for detecting *XPO1*^E571K^ mutation in cfDNA from peripheral blood of lymphoma patients. By combining the high specificity of allele-specific amplification with the enhanced sensitivity of temperature-optimized PCR conditions, this approach aims to provide a cost-effective, accessible, and highly sensitive platform for lymphoma screening and monitoring.

## 2. Results

### 2.1. Network Analysis of XPO1 in Lymphoma

To explore and emphasize the functional role of XPO1 in lymphoma, Kyoto Encyclopedia of Genes and Genomes (KEGG) pathway enrichment, Gene Ontology (GO) analysis, and protein–protein interaction (PPI) network analysis were performed. KEGG pathway analysis revealed that XPO1 is associated with key lymphoma-related pathways, including the pathway in cancer, the PI3K-Akt signaling pathway, and the JAK-STAT signaling pathway, all of which regulate cancer cell survival, proliferation, and immune evasion (Figure 1A). GO analysis further demonstrated that XPO1 is involved in critical biological processes essential for lymphoma progression. Cellular component (CC) analysis highlighted the localization of XPO1 within the nucleus, cytoplasm, and extracellular exosome, while molecular function (MF) analysis identified its role in protein binding and RNA binding, further supporting its regulatory function in tumor progression (Figure 1B) [27]. PPI network analysis using STRING and visualization via Cytoscape (version 3.10.3, available at https://cytoscape.org/, accessed on 1 March 2025), identified XPO1 as a central hub, interacting with key lymphoma-associated proteins (Figure 2). These findings suggest that XPO1 plays a pivotal role in lymphoma by modulating multiple oncogenic pathways and protein interactions.

### 2.2. Assessment of Plasma and FFPE Tissue Samples for XPO1 Gene Amplification and Sequencing in Lymphoma Patients

Thirty peripheral blood samples and twenty-four formalin-fixed paraffin-embedded (FFPE) tissue samples were collected from lymphoma patients diagnosed at the Central Laboratory of Maharaj Nakorn in Chiang Mai province, Thailand. Following collection, all samples underwent amplification via conventional PCR assay, followed by analysis using 1.5% agarose gel electrophoresis. The Hemoglobin Subunit Beta (*HBB*) gene served as an internal control to validate the successful extraction of both genomic DNA and cfDNA from plasma samples. Notably, bands were observed on the agarose gel for PCR products from all plasma samples (Figure 3A). Subsequently, the XPO1-targeted gene was amplified, and all resulting PCR products were forwarded to ATGC Co., Ltd., Bangkok, Thailand. Direct sequencing was performed using ABI 3730xl DNA sequencers (Thermo Fisher Scientific, Waltham, MA, USA). Unfortunately, interpretation of all obtained sequences of plasma DNA proved challenging, which may be attributed to the heterogeneity of the PCR product (130 bp), as shown in Figure 3B, rather than solely the short amplicon length.

### 2.3. Optimal Critical Temperature (Tc) of FAST-COLD-PCR Assay and Verification of FAST-COLD-PCR Assay for XPO1^E571K^ Detection

The optimal critical temperature (Tc) for the FAST-COLD-PCR assay was determined through systematic evaluation to maximize the enrichment of mutant product amplification while suppressing wild-type product generation. This optimization utilized synthesized *XPO1^E571K^* single-strand DNA (ssDNA) fragments representing the mutated *XPO1* gene and cell line (Raw 246.7) samples representing the wild-type *XPO1*. First, the products were subjected to high-resolution melt (HRM) analysis, which generated melting profiles in the form of HRM difference curves. The results indicated that the average melting temperature (Tm) of *XPO1*^E571K^ was 75.0 °C, while the wild type exhibited a Tm of 75.5 °C (Figure 4), demonstrating that the wild type possessed a higher melting temperature compared with the mutant variant. Based on these thermal characteristics, the FAST-COLD-PCR assay was selected for mutation detection. Tc was determined to identify the precise thermal conditions for enriching the mutant without amplifying the wild type. Subsequently, Tc served as the denaturation temperature following 10 cycles of PCR amplification, aiming to amplify the template of both wild type and mutants before implementing selective thermal conditions. Tc values ranged from 74.0 to 72.4 °C, facilitating selective denaturation of the mutant sequence for amplification while preventing wild-type amplification. PCR products (130 bp) were subsequently analyzed using 1.5% agarose gel electrophoresis. The results demonstrated that at Tc = 73.3 °C, selective denaturation and amplification of mutant products occurred without detectable wild-type product formation (Figure 5). Thus, 73.3 °C was established as the optimal Tc for detecting *XPO1*^E571K^ mutation in the FAST-COLD-PCR assay. Complete PCR reaction conditions are detailed in Table 1.

### 2.4. Limit of Detection and of FAST-COLD-PCR Assay for XPO1^E571K^ Mutation Detection

The limit of detection (LOD) for the FAST-COLD-PCR assay represents the minimum concentration of DNA template required to reliably detect mutations in the *XPO1* gene. In this study, 2.4 ng of synthesized *XPO1^E571K^* ssDNA fragments (undiluted sample; UD) were serially diluted into 24 ng of patient-derived DNA samples at ratios of 1:10 (240 pg), 1:50 (48 pg), and 1:100 (24 pg). A patient-derived sample containing only wild-type *XPO1* sequences served as the negative control, while synthesized *XPO1^E571K^* ssDNA fragments were used as the positive control. Additionally, no-template control (NTC) was included to ensure PCR system integrity and rule out contamination. PCR amplifications were performed according to previously described conditions of the FAST-COLD-PCR, and products were analyzed by 1.5% agarose gel electrophoresis. The LOD was determined based on the presence of visible bands observed by the naked eye. The lowest detectable concentration of synthesized *XPO1*^E571K^ ssDNA fragments was at the 1:100 dilution, corresponding to approximately 24 pg of mutated DNA template within patient-derived DNA sample (Figure 6).

### 2.5. Detecting XPO1^E571K^ Gene Mutation in Patient Samples and FFPE Tissue Samples Using the FAST-COLD-PCR Assay

Following the optimization of the FAST-COLD-PCR assay with a critical temperature (Tc) set at 73.3 °C, genomic DNA and cfDNA were extracted from 30 peripheral blood plasma samples and 24 FFPE tissue samples collected from lymphoma patients. All samples were then subjected to analysis using the optimized FAST-COLD-PCR conditions. Remarkably, the *XPO1*^E571K^ mutation was detected in 8 out of 30 plasma samples, yielding a detection rate of 26.7% (Figure 7, Table 2). In contrast, none of the 24 FFPE tissue samples tested positive for the *XPO1*^E571K^ mutation (Figure 8, Table 3). Importantly, all FAST-COLD-PCR results were fully validated by direct sequencing, confirming complete concordance and underscoring the reliability and accuracy of the assay for *XPO1*^E571K^ mutation detection in clinical specimens.

## 3. Discussion

*XPO1* or *CRM1* gene encodes the exportin 1 protein, a critical nuclear export factor responsible for transporting cargo proteins and ribonucleoproteins from the nucleus to the cytoplasm in PMBL and cHL. The present study demonstrated that mutations in the *XPO1* gene result in alterations at the molecular level in cancer cells, particularly in lymphoid malignancies. Bioinformatic network analysis provided comprehensive evidence for XPO1’s involvement in lymphomagenesis. KEGG pathway analysis revealed that XPO1 is associated with key lymphoma-related pathways, including the pathway in cancer, the PI3K-Akt signaling pathway, and the JAK-STAT signaling pathway, all of which are fundamental regulators of cancer cell survival, proliferation, and immune evasion [28,29]. Correspondingly, GO analysis further demonstrated XPO1’s participation in essential biological processes that reinforce its pivotal role in lymphoma progression. Additionally, PPI network analysis identified XPO1 as a central hub protein with extensive interactions involving key lymphoma-associated regulators, such as BCL2, TP53, and MYC, which are known regulators of apoptosis and tumor progression [30]. This central positioning within critical oncogenic networks underscores the clinical significance of detecting XPO1 mutations in lymphoma diagnosis and monitoring. Given XPO1’s integral role in these interconnected oncogenic pathways and its interactions with established tumor suppressors and oncogenes, the detection of XPO1 mutations represents a crucial component of comprehensive lymphoma molecular profiling.

The *XPO1*^E571K^ mutation is frequently observed in lymphoma patients and represents a clinically significant genetic alteration [9]. Importantly, the presence of the *XPO1*^E571K^ mutation is associated with favorable prognosis in lymphoma patients [10] as this mutation alters nuclear export signal recognition in a sequence-specific manner and enhances cellular sensitivity to therapeutic agents targeting XPO1 protein function currently under clinical development [31]. Moreover, *XPO1*^E571K^ was reported to render lymphoma cells more sensitive to Selinexor, a selective inhibitor of nuclear export utilized as a chemotherapeutic agent. This enhanced sensitivity results from accelerated degradation of the mutant XPO1 protein compared with the wild-type variant [32]. This enhanced sensitivity results from accelerated degradation of the mutant XPO1 protein compared with the wild-type variant. Recent studies have evaluated the efficacy of a first-in-class HSP110 inhibitor (iHSP110-33) both as a single agent and in combination with Selinexor. The results demonstrated that iHSP110-33 treatment significantly reduced the survival of multiple PMBL and cHL cell lines, as well as decreased tumor xenograft size. Moreover, the combination of iHSP110-33/Selinexor could induce a synergistic reduction in STAT6 phosphorylation and lymphoma cell growth in vitro and in vivo [33]. Therefore, detection of gene mutation is important for risk assessment and the decision-making of doctors in the planning of lymphoma treatment. Moreover, efficiency and specificity of the method for *XPO1*^E571K^ cfDNA detection are very important as well. Nowadays, there are several methods to detect *XPO1*^E571K^ gene mutation. However, these approaches present significant limitations. Existing methods are characterized by high costs and complex procedural requirements. Furthermore, result interpretation demands specialized expertise and requires sophisticated instrumentation for experimental execution. Most importantly, conventional detection methods demonstrate limited sensitivity for detecting low-abundance gene mutations, which represents a critical limitation in clinical laboratory settings where early detection and monitoring of minimal residual diseases are essential. These technical and practical constraints highlight the need for improved detection methodologies that can overcome the limitations of current approaches while maintaining high specificity and sensitivity for *XPO1*^E571K^ mutation detection in clinical specimens.

Normally, gene mutation is always examined from DNA extracted from tumor tissues. This process is an invasive technique. Thus, a molecular target released from cancer cells and suspended in the blood circulation of patients, which is called cell-free circulating tumor DNA (cfDNA), is interesting. However, levels of cfDNA concentration are characteristically low in cancer patients, presenting significant analytical challenges. Therefore, the technique used for detection should be of high specificity and sensitivity to reliably identify mutations in these dilute samples. To address these technical limitations, we developed a technique for detecting the *XPO1*^E571K^ gene mutations in lymphoma patients with high accuracy, specificity, and sensitivity with rapid turnaround time, while maintaining low procedural complexity and cost-effectiveness. This technique, designated FAST-COLD-PCR, represents a targeted approach specifically designed to overcome the sensitivity limitations inherent in conventional mutation detection methods when applied to low-abundance cfDNA samples. The results demonstrated that FAST-COLD-PCR is highly suitable for detecting *XPO1*^E571K^ gene mutation in cfDNA from plasma of lymphoma patients, with an optimal critical temperature (Tc) of 73.3 °C. This Tc value effectively suppressed wild-type DNA amplification (no detectable bands) while selectively enriching mutated gene copies following FAST-COLD-PCR amplification.

This selective amplification was further confirmed by spiking *XPO1*^E571K^ ssDNA mutation fragments into a patient-derived sample, thereby simulating the complexity of actual DNA extracts containing various gene fragments. These results demonstrated that the FAST-COLD-PCR conditions can efficiently and specifically amplify *XPO1*^E571K^ mutations fragments in the presence of an excess of background DNA. Additionally, the limit of the detection was determined to be approximately 24 pg, which is below the typical cfDNA yield that generally exceeds 20 ng, highlighting the method’s suitability for liquid biopsy applications. In addition to its high sensitivity and specificity, FAST-COLD-PCR is cost-effective, rapid, and easily integrated into standard laboratory workflows without the need for specialized equipment. Its optimized conditions reduce false positives and allow detection of low-frequency mutations, making it an excellent tool for non-invasive cancer diagnostics and real-time patient monitoring.

Consequently, *XPO1*^E571K^ gene mutations could be detected directly in plasma samples from lymphoma patients. This study demonstrated that 26.7% of peripheral blood samples harbored *XPO1*^E571K^ gene mutations. Notably, the *XPO1*^E571K^ gene mutation could not be found in the FFPE tissue samples collected from lymphoma patients. These findings are consistent with the previous study in which *XPO1* gene mutations are detectable in plasma but not in tumor tissue, attributed to the characteristic tumor cell sparsity observed in lymphoma [10]. However, analysis of clinical correlations revealed that cfDNA containing *XPO1^E571K^* gene mutations showed no significant associations with patient demographics (sex or age), lymphoma subtype, or clinical status. The study was constrained by a limited sample size due to the relatively low incidence of lymphoma cases at the participating institution. However, sample size calculation using n4Studies software version 1.3.0 confirmed that 15 samples were statistically adequate for this investigation. This study establishes the methodology and optimal conditions for *XPO1*^E571K^ gene detection in lymphoma patients using plasma-based analysis. The findings demonstrate that FAST-COLD-PCR represents a viable approach for clinical laboratory implementation, although it requires access to sophisticated laboratory technology and specialized instrumentation. The method’s ability to detect mutations in cfDNA while avoiding the limitations associated with tissue-based analysis makes it particularly valuable for lymphoma molecular diagnostics.

In conclusion, this study successfully established and evaluated a FAST-COLD-PCR assay for the sensitive detection *of XPO1*^E571K^ mutations in patient-derived samples. Standard PCR analysis of peripheral blood samples revealed challenges in sequence interpretation due to product length limitations. By determining the optimal critical temperature (Tc) at 73.3 °C, FAST-COLD-PCR enabled the selective amplification of mutant *XPO1*^E571K^ alleles while suppressing wild-type sequences. The assay demonstrated a limit of detection of approximately 24 pg of mutant DNA in a background of wild-type genomic DNA. This method proved effective in detecting *XPO1*^E571K^ mutations in peripheral blood samples, with results concordant with direct sequencing. Overall, the developed FAST-COLD-PCR assay presents a sensitive, rapid, and cost-effective approach for the non-invasive detection of *XPO1*^E571K^ mutations in plasma, underscoring its potential clinical utility in lymphoma management. Further validation using larger patient cohorts is recommended to confirm these findings and establish the assay’s clinical applicability.

## 4. Materials and Methods

### 4.1. Bioinformatics Analysis

Gene Ontology (GO) term and Kyoto Encyclopedia of Genes and Genomes (KEGG) pathway enrichment analyses were conducted to investigate the biological significance of *XPO1* in lymphoma [34]. Genes associated with lymphoma, including *XPO1*, were retrieved from the GeneCards database (https://www.genecards.org, accessed on 1 March 2025) by searching the term “Lymphoma”. The resulting gene set was then submitted to DAVID (https://david.ncifcrf.gov/, accessed on 1 March 2025) for functional annotation and enrichment analysis [35,36]. GO analysis was performed to categorize lymphoma-associated genes into three main categories: (1) biological processes (BPs), (2) cellular component (CCs), and (3) molecular functions (MFs). Pathways relevant to lymphoma were identified in the KEGG database based on their association in cancer signaling, immune modulation, and apoptosis. Visualization of the enrichment results was performed using SRplot (https://www.bioinformatics.com.cn/srplot, accessed on 1 March 2025), emphasizing significantly enriched terms according to *p*-values (−log_10_ transformed) [37].

Protein–protein interaction (PPI) network analysis was conducted using STRING (version 12.0, available at https://string-db.org/, accessed on 1 March 2025) to further explore the role of *XPO1* in lymphoma. Cytoscape (version 3.10.3, available at https://cytoscape.org/, accessed on 1 March 2025) was used for network visualization and topological analysis [38]. Proteins involved in critical regulatory pathways in lymphoma, such as *XPO1*, *BCL2*, and *TP53*, were identified among the key interactors.

### 4.2. Patients and Sample Collections

Thirty peripheral EDTA blood samples and twenty-four FFPE tissue samples were obtained from patients diagnosed with lymphoma at the Central Laboratory of Maharaj Nakorn Chiang Mai, Faculty of Medicine, Chiang Mai University, Chiang Mai, Thailand. Sample collection was performed between October 2022 and November 2024. The sample size was calculated using a finite population proportion formula by free application (n4Studies). This study was approved by the Ethics Committee of the Faculty of Medicine, Chiang Mai University, Chiang Mai, Thailand. The study code was NONE-2565-09230/Research ID 9230; the date of approval was 17 October 2022.

### 4.3. Genomic DNA and cfDNA Preparations

Plasma DNA samples were extracted from peripheral EDTA blood samples obtained from lymphoma patients using Purelink^™^ Genomic DNA Mini Kit (Life Technologies Corp., Carlsbad, CA, USA) according to the manufacturer’s protocol. Genomic DNA was obtained from FFPE tissue samples of the lymphoma patients using QIAamp^®^ DNA FFPE Advanced Kit (Qiagen, Shanghai, China) according to the manufacturer’s protocols. The concentration of DNA was determined using the Qubit DNA Assay kit (Invitrogen, Thermo Fisher Scientific, Waltham, MA, USA) and Qubit 4 fluorometer (Invitrogen, Thermo Fisher Scientific, Waltham, MA, USA) and kept at −80 °C until use.

### 4.4. Detection of XPO1^E571K^ in Patient Samples and FFPE Tissue Samples Using Conventional PCR Assay

Conventional PCR assay was performed on the plasma DNA samples and genomic DNA obtained from 30 lymphoma patients and 24 FFPE tissue samples to detect *XPO1*^E571K^ mutation. The PCR reaction was prepared at the final volume of 20 µL containing 2× Quick Taq HS DyeMix (Toyobo, Tokyo, Japan), 0.2 µM of each *XPO1* forward primer (5′-CTGGAAATTTCTGAAGACTGTAGTT-3′) and reverse primer (5′-GGGTCTCTAACAA GACAAAAACA-3′), and approximately 1.0 × 10^−2^ ng of DNA. DNA extracted from Raw 264.7 cell line and synthesized *XPO1*^E571K^ ssDNA fragments (360 nucleotides) were used as the wild type and mutated control, respectively. PCR was performed in a CFX96 Touch Real-Time PCR System (Bio-Rad Laboratories, Inc., Hercules, CA, USA), including initial denaturation 94 °C for 3 min, followed by 35 cycles at 94 °C for 30 s, 60 °C for 30 s, and 72 °C for 30 s, with a final extension at 72 °C for 3 min. PCR products were analyzed by 1.5% agarose gel electrophoresis with running condition at 10 volts/cm and visualized using the Gel Doc XR+ System (Bio-Rad Laboratories, Inc., Singapore). All PCR products were verified by Sanger sequencing (ATGC Co., Ltd., Bangkok, Thailand).

### 4.5. Development of FAST-COLD-PCR Assay

The Tc was determined to selectively denature only the mutant–mutant homoduplexes, while keeping the wild-type–wild-type homoduplexes intact. The temperature was chosen to allow specific amplification of the mutant product without enhancing the amplification of the wild-type product. To determine the Tc of the FAST-COLD-PCR assay, PCR reaction was prepared at a final volume of 20 µL containing 1× SensiFAST HRM Master Mix (Meridian Bioscience, Inc., Cincinnati, OH, USA), 0.2 µM of *XPO1* forward primer (5′-CTGGAAATTTCTGAAGACTGTAGTT-3′) and reverse primer (5′-GGGTCTCTAACAAGACAAAAACA-3′) and approximately 10 ng of Raw 264.7 cell line DNA or 1.0 × 10^−2^ ng of synthesized *XPO1*^E571K^ ssDNA fragments. PCR was performed using the CFX96 Touch Real-Time PCR System (Bio-Rad Laboratories, Inc., Hercules, CA, USA), including initial denaturation at 94 °C for 3 min, followed by three cycles at 94 °C for 5 s, 60 °C for 10 s, and 72 °C for 30 s, followed by 40 cycles of denaturation at 94 °C for 5 s, various Tc ranges (80–90 °C) for 40 s, primer annealing at 60 °C for 10 s, and extension at 72 °C for 30 s, with a final extension at 72 °C for 3 min. The final step was a melting curve with a continuous increase in temperature from 65 °C to 95 °C (0.2 °C per acquisition and 5 s holds before each acquisition). Finally, HRM was analyzed using Precision Melt Analysis™ software version 1.3 for differentiating wild-type and mutant alleles.

After determination of the Tc, the FAST-COLD-PCR assay was used to verify the enrichment of the detection of *XPO1*^E571K^ mutations. In brief, the PCR reaction was prepared at a final volume of 20 µL containing 2× Quick Taq HS DyeMix (Toyobo, Japan), 0.2 µM of each primer, and approximately 10 ng of Raw 264.7 cell line DNA or 10^−2^ ng of synthesized *XPO1*^E571K^ ssDNA fragments. PCR was performed in a Mini MJ Thermal Cycler (Bio-Rad Laboratories Inc., Singapore), including initial denaturation at 94 °C for 2 min; followed by ten cycles at 94 °C for 30 s, 60 °C for 30 s, and 68 °C for 30 s; and then followed by 30 cycles of critical denaturation at 73.3 °C for 30 s, primer annealing at 56 °C for 30 s, and extension at 68 °C for 30 s, with a final extension at 68 °C for 3 min. PCR products were analyzed by 1.5% agarose gel electrophoresis with running condition at 10 volts/cm and visualized using the Gel Doc XR+ System (Bio-Rad Laboratories Inc., Singapore).

### 4.6. Assesment of the Limit of Detection of FAST-COLD-PCR Assay

Following the development of the FAST-COLD-PCR assay, the limit of detection for *XPO1*^E571K^ mutations was evaluated by spiking mutation DNA fragments into patient-derived samples to mimic the composition of actual DNA extracts. In brief, PCR reaction was prepared at a final volume of 20 µL, containing 2× Quick Taq HS DyeMix (Toyobo, Japan), 0.2 µM of each primer, and varying amounts of DNA templates under different conditions. To assess the limit of detection, 2.4 ng of synthesized *XPO1*^E571K^ ssDNA fragments were serially diluted into 24 ng of patient-derived sample containing the wild-type XPO1 at ratios of 1:10, 1:50, and 1:100. A patient-derived sample, previously confirmed to carry only the wild-type XPO1 sequence by Sanger sequencing (ATGC Co., Ltd., Bangkok, Thailand), was used as a negative control. The positive control consisted of 2.4 ng of synthesized *XPO1*^E571K^ ssDNA fragments. No-template controls (NTCs) were included in each PCR run to monitor for system integrity and to exclude potential contamination. PCR amplification was performed using the previously optimized FAST-COLD-PCR conditions. The PCR products were resolved by electrophoresis on a 1.5% agarose gel and visualized using the Gel Doc XR+ System (Bio-Rad Laboratories Inc., Singapore).

## Figures and Tables

**Figure 1 ijms-26-07324-f001:**
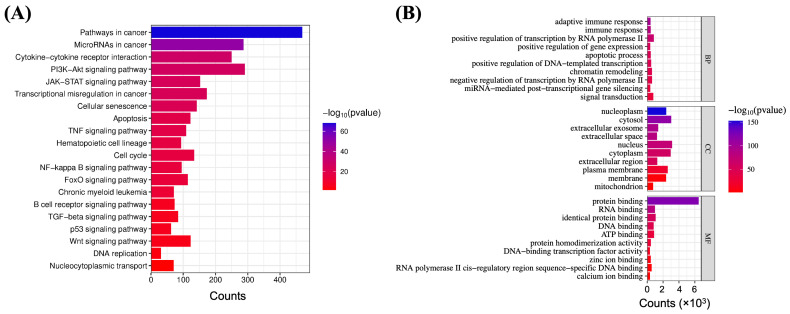
(**A**) KEGG pathway and (**B**) Gene Ontology (GO) enrichment analysis of lymphoma. KEGG analysis identified key lymphoma-associated pathways. GO analysis categorized genes involved in lymphoma into biological processes (BPs), cellular components (CCs), and molecular functions (MFs), highlighting their involvement in processes contributing to lymphoma progression.

**Figure 2 ijms-26-07324-f002:**
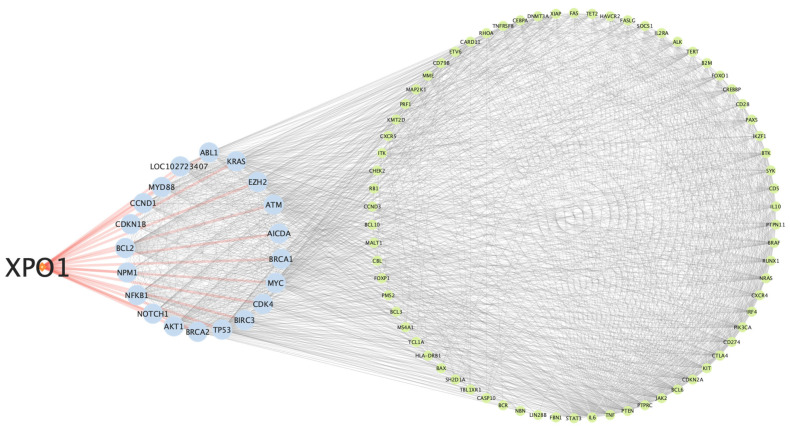
Protein–protein interaction (PPI) network analysis of XPO1. The PPI network, constructed using STRING and visualized with Cytoscape, identifies XPO1 as a central hub interacting with key lymphoma-associated proteins.

**Figure 3 ijms-26-07324-f003:**
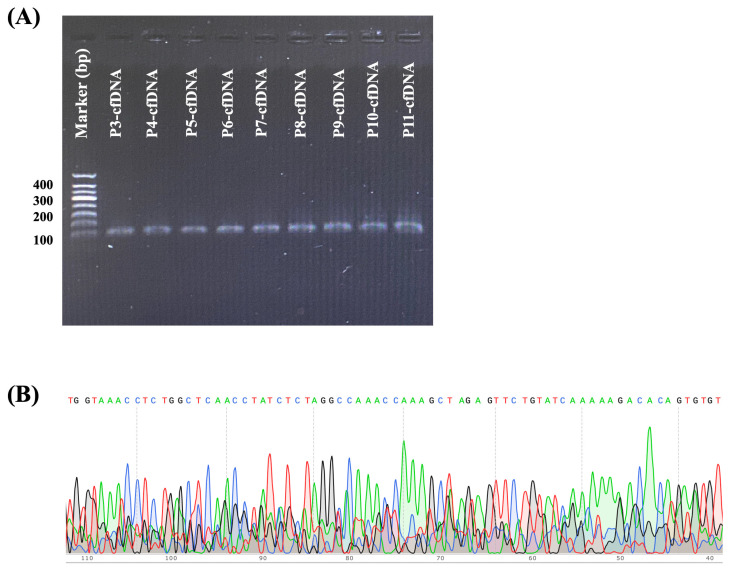
(**A**) Representative *HBB* gene bands of plasma samples from lymphoma patients, serving as a housekeeping gene to confirm successful DNA extraction. (**B**) Example of direct sequencing in the reverse direction of PCR products, which yielded uninterpretable results.

**Figure 4 ijms-26-07324-f004:**
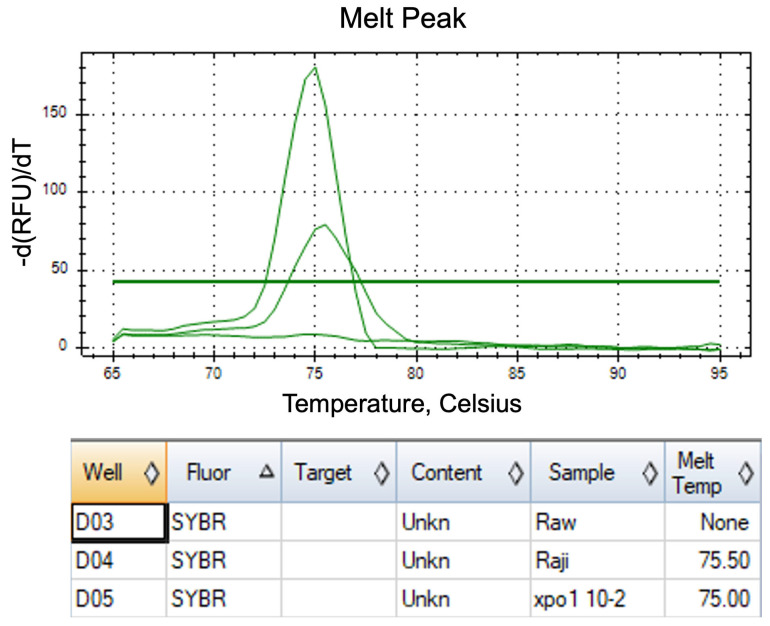
Tm of wild-type and mutant PCR products. Wild type (Raji) shows a melting temperature of 75.5 °C while the mutant (xpo1 10^−2^) exhibits a lower Tm of 75.0 °C. Raw represents the negative control with no detectable melting peak. Green lines indicate reference thresholds for melt peak analysis.

**Figure 5 ijms-26-07324-f005:**
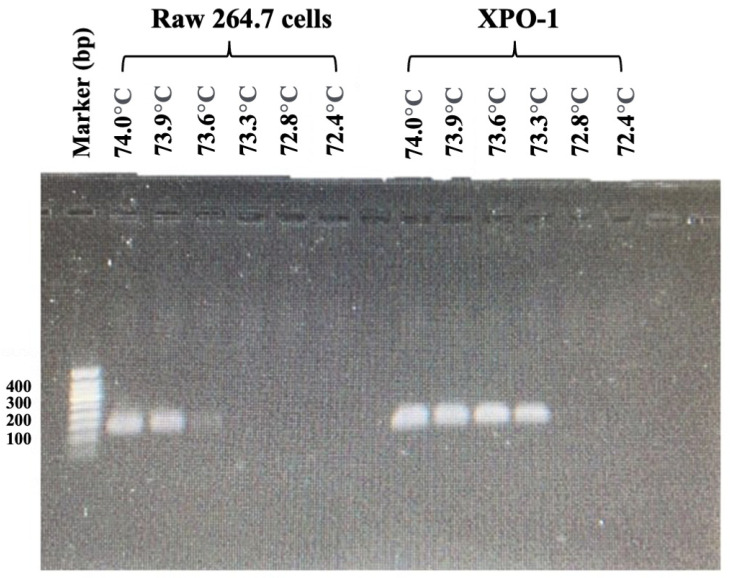
Tc temperature optimization for the FAST-COLD-PCR assay varied between 74.0 and 72.4 °C. The experiment was repeated three times with consistent results.

**Figure 6 ijms-26-07324-f006:**
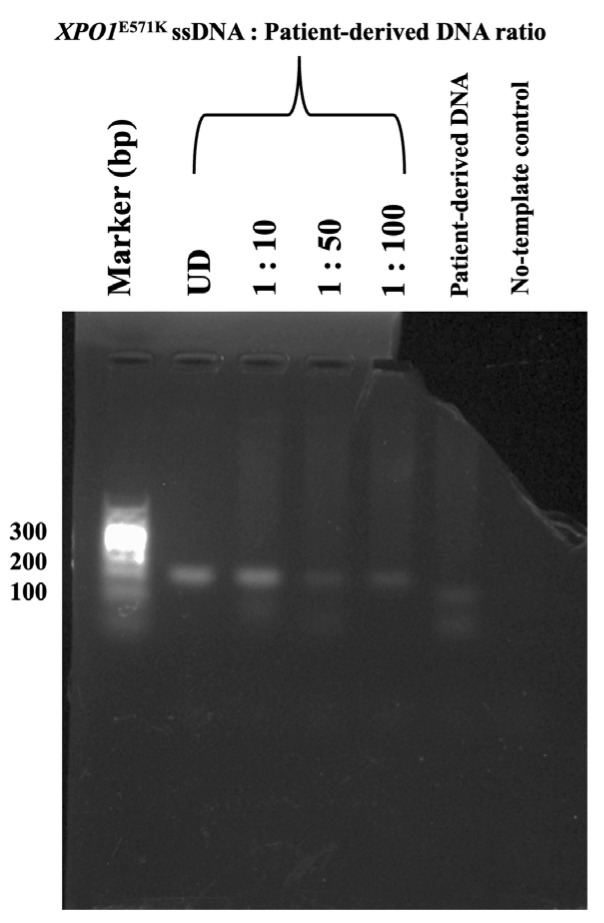
Limit of detection (LOD) of *XPO1*^E571K^ mutation using FAST-COLD-PCR. Synthesized XPO1^E571K^ ssDNA fragments (2.4 ng; undiluted sample or UD) were serially diluted into patient-derived DNA samples at ratios of 1:10 (240 pg), 1:50 (48 pg), and 1:100 (24 pg) to simulate actual samples. The mutation was detectable to the 1:100 dilution. No bands were observed in the patient-derived DNA samples containing only wild-type XPO1 sequences or in the no-template control. The experiment was repeated three times with consistent results.

**Figure 7 ijms-26-07324-f007:**
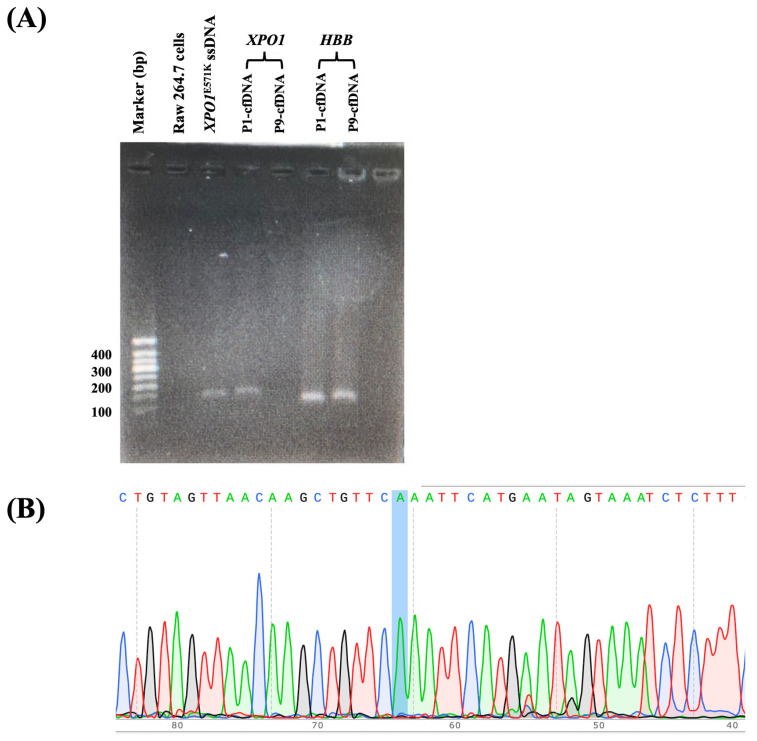
Example of peripheral blood samples collected from lymphoma patients who underwent FAST-COLD-PCR assay. (**A**) P1-cfDNA showed a distinct band, confirmed by its direct sequencing to have an *XPO1* gene mutation status. The *HBB* gene served as an internal control to validate the successful extraction of cfDNA from plasma samples. The position of the *XPO1^E571K^* point mutation is indicated by a blue shadow. (**B**) Direct sequencing confirmed mutant status of P1-cfDNA.

**Figure 8 ijms-26-07324-f008:**
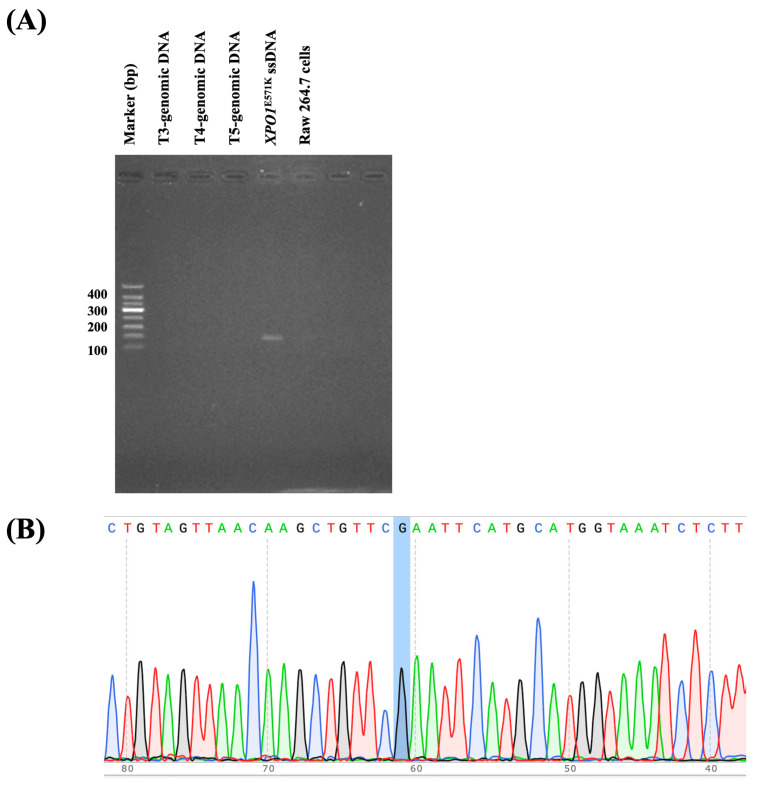
Example of FFPE tissue samples collected from lymphoma patients that underwent FAST-COLD-PCR assay. (**A**) All samples showed no bands. (**B**) Direct sequencing confirmed wild-type status of all samples. The position of the *XPO1^E571K^* point mutation is indicated by a blue shadow.

**Table 1 ijms-26-07324-t001:** FAST-COLD-PCR reaction condition.

Cycle Step	Temp	Time	Cycles
Initial denaturation	94.0 °C	2 min	1
Denaturation	94.0 °C	30 s	10
Annealing	60.0 °C	30 s	
Extension	68.0 °C	30 s	
Denaturation (Tc)	73.3 °C	30 s	30
Annealing	60.0 °C	30 s	
Extension	68.0 °C	30 s	
Hold	4.0 °C	∞	

**Table 2 ijms-26-07324-t002:** *XPO1*^E571K^ mutation status from cfDNA of patient samples having undergone FAST-COLD-PCR assay.

Sample (Patient No.)	Sex	Age	Lymphoma Type	Patient Status	*XPO1*^E571K^ Mutation
1	Female	67	DLBCL	New	+
2	Female	54	DLBCL	Refractory	+
3	Female	64	DLBCL	Refractory	+
4	Male	70	BCL	Refractory	+
5	Female	71	NHL	New	+
6	Female	71	NHL	Refractory	+
7	Male	61	DLBCL	Refractory	+
8	Female	69	DLBCL	Refractory	+
9	Female	20	NHL	Refractory	−
10	Female	NA	N/A	N/A	−
11	Male	70	NHL	N/A	−
12	Female	N/A	N/A	N/A	−
13	Male	N/A	N/A	N/A	−
14	Male	N/A	N/A	N/A	−
15	Female	20	NHL	N/A	−
16	Male	73	N/A	N/A	−
17	Male	29	NHL	N/A	−
18	Male	58	N/A	N/A	−
19	Male	28	HL	Refractory	−
20	Female	N/A	BCL	N/A	−
21	Female	51	BCL	N/A	−
22	Male	60	HL	N/A	−
23	Female	53	NHL	N/A	−
24	Female	32	NHL	N/A	−
25	Male	40	N/A	N/A	−
26	Female	21	HL	N/A	−
27	Male	N/A	NHL	N/A	−
28	Female	64	DLBCL	New	−
29	Female	N/A	N/A	N/A	−
30	Female	55	N/A	N/A	−

BCL, B-cell lymphoma; DLBCL, diffuse large B-cell lymphoma; HL, Hodgkin lymphoma; NHL, non-Hodgkin lymphoma; N/A = not assessed.

**Table 3 ijms-26-07324-t003:** *XPO1*^E571K^ mutation status from FFPE samples having undergone FAST-COLD-PCR assay.

Sample (Patient No.)	Sex	Age	Lymphoma Type	Patient Status	*XPO1*^E571K^ Mutation
1	Male	44	DLBCL	Refractory	−
2	Female	54	DLBCL	Refractory	−
3	Male	53	DLBCL	New	−
4	Male	56	DLBCL	New	−
5	Male	21	DLBCL	New	−
6	Male	59	DLBCL	New	−
7	Female	44	DLBCL	New	−
8	Female	55	DLBCL	New	−
9	Female	64	DLBCL	New	−
10	Male	73	DLBCL	New	−
11	Male	75	DLBCL	Refractory	−
12	Female	66	DLBCL	Refractory	−
13	Male	74	DLBCL	New	−
14	Female	68	DLBCL	New	−
15	Male	68	DLBCL	New	−
16	Female	64	NHL	Relapsed	−
17	Male	66	DLBCL	Relapsed	−
18	Female	68	DLBCL	Relapsed	−
19	Female	80	DLBCL	New	−
20	Male	75	DLBCL	New	−
21	Female	71	DLBCL	New	−
22	Female	68	DLBCL	New	−
23	Male	66	DLBCL	New	−
24	Female	64	DLBCL	New	−

DLBCL, diffuse large B-cell lymphoma; NHL, non-Hodgkin lymphoma.

## Data Availability

The original contributions presented in the study are included in the article; further inquiries can be directed to the corresponding authors.

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
