# Peer review of "Detection of XPO1E571K Gene Mutation from Cell-Free DNA in Blood Circulation of Lymphoma Patients by FAST-COLD PCR"

_ijms, 2025, doi:10.3390/ijms26157324_

Round 1
Reviewer 1 Report
Comments and Suggestions for Authors
The paper is solely dedicated to the optimization of the PCR conditions to detect specifically a mutated version of the gene XPO1 in human samples. I understand a clinical value of this work, although scientifically speaking it is not a breakthrough. Such works were popular approximately 20 years ago. But it is up to the editors to decide whether the scientific value of this work is sufficiently high. The introduction is very detailed about the lymphomas in general, but I miss a part dedicated to a various tricks and alternative approaches to discriminate between wild type and mutated isoforms of particular genes. I would suggest to make an introduction a bit more focused on the subject, rather than general. I fully appreciate the clinical importance of the method, it might be useful indeed. Could authors describe a bit more in details alternative methods used in the clinic for this and similar cases?
Some small details. Fig3 indicates heterogeneity of the PCR product, but authors for some reason interpret their difficulties due to a short length of it. (Paragraph above). Please correct.
At last I do not comprehend why authors suggest the name FAST COLD, whereas they get specificity by using higher annealing temperature for the mutant form compared to the WT.
Author Response
Reviewer 1
1. The paper is solely dedicated to the optimization of the PCR conditions to detect specifically a mutated version of the gene XPO1 in human samples. I understand a clinical value of this work, although scientifically speaking it is not a breakthrough. Such works were popular approximately 20 years ago. But it is up to the editors to decide whether the scientific value of this work is sufficiently high. The introduction is very detailed about the lymphomas in general, but I miss a part dedicated to a various tricks and alternative approaches to discriminate between wild type and mutated isoforms of particular genes. I would suggest to make an introduction a bit more focused on the subject, rather than general. I fully appreciate the clinical importance of the method, it might be useful indeed. Could authors describe a bit more in details alternative methods used in the clinic for this and similar cases?
Response: Thank you for this insightful suggestion. We have revised the Introduction to more clearly focus on mutation detection techniques, particularly in the context of distinguishing between wild-type and mutated gene isoforms such as XPO1E571K. We also added a comparative overview of alternative molecular methods currently used in clinical and research settings, including allele-specific PCR, TaqMan assays, digital PCR (dPCR), droplet digital PCR (ddPCR), and NGS platforms. These additions can be found on pages 3–4, lines 133–151.
2. Some small details. Fig3 indicates heterogeneity of the PCR product, but authors for some reason interpret their difficulties due to a short length of it. (Paragraph above). Please correct.
Response: Thank you for your valuable comment. We appreciate the reviewer’s observation. In the revised manuscript, we have corrected the interpretation of the unclear sequencing results shown in Figure 3B. The potential contribution of PCR product heterogeneity-likely caused by non-specific amplification or secondary structure formation-is now addressed, rather than attributing the issue solely to the short amplicon length. This revision appears on pages 6–7, lines 208–211.
3. At last I do not comprehend why authors suggest the name FAST COLD, whereas they get specificity by using higher annealing temperature for the mutant form compared to the WT.
Response: Thank you for pointing out the inconsistency in melting temperature interpretation. We have corrected the error in the manuscript. As shown in Figure 4, the XPO1E571K mutant has a lower melting temperature (75.0°C) compared to the wild-type (75.5°C), classifying it as a Tm-reducing mutation. We have clarified that our use of the term "FAST-COLD-PCR" aligns with existing COLD-PCR variants in the literature. In particular, fast-COLD-PCR selectively enriches low–melting temperature alleles through a critical denaturation temperature (Tc) that favors mutant allele amplification. This clarification has been added on page 6, lines 226–227.
Reviewer 2 Report
Comments and Suggestions for Authors
This study by Duangmano et al. employed FAST-COLD PCR to detect XPO1E571K from cfDNA circulating in the blood of lymphoma patients. This approach shows promise as a potential tool for invasive clinical screening in lymphoma. Below are specific comments:
- For all KEGG and GO analyses, it is unclear where the gene sets originate. Are they based on differentially expressed genes between lymphoma patients and healthy donors? Please provide more details regarding the source and selection criteria of these genes.
- In Figure 6, if XPO1E571K mutant ssDNA fragments exist at even lower concentrations, would the method still be sensitive enough to detect them?
- The overall detection rate (26.7%) appears relatively low. Could this be due to the low abundance of ssDNA fragments, or is it related to variation in mutation-specific annealing temperatures? Further explanation would be helpful.
- For most experiments, the number of technical or biological replicates is not specified. Please include this information in the figure legends.
Author Response
Reviewer 2
This study by Duangmano et al. employed FAST-COLD PCR to detect XPO1E571K from cfDNA circulating in the blood of lymphoma patients. This approach shows promise as a potential tool for invasive clinical screening in lymphoma. Below are specific comments:
1. For all KEGG and GO analyses, it is unclear where the gene sets originate. Are they based on differentially expressed genes between lymphoma patients and healthy donors? Please provide more details regarding the source and selection criteria of these genes.
Response: We thank the reviewer for this important comment. The gene sets used for GO and KEGG enrichment analyses were not based on differentially expressed genes between lymphoma patients and healthy donors. Instead, we retrieved genes associated with lymphoma from the GeneCards database (https://www.genecards.org) by searching the term “lymphoma.” This gene set, which includes XPO1, was then submitted to the DAVID functional annotation tool for GO and KEGG pathway enrichment analysis. The results of the Go and KEGG enrichment analyses were visualized based on gene counts. We have now clarified this in more detail in the Methods section of the manuscript with yellow highlight on page 14, lines 418-435.
2. In Figure 6, if XPO1E571K mutant ssDNA fragments exist at even lower concentrations, would the method still be sensitive enough to detect them?
Response: Based on our serial dilution experiments in Figure 6, our FAST-COLD PCR method demonstrates reliable detection of XPO1E571K mutations down to 1:100 dilution, which corresponds to approximately 1–0.1% mutant allele frequency. While our current sensitivity range is clinically relevant for most lymphoma cfDNA applications where tumor-derived DNA typically represents 0.1–10% of total circulating DNA, future improvements could enhance ultra-low detection through strategies such as increasing input DNA volumes, optimizing PCR parameters, implementing digital PCR approaches, or using more sensitive real-time detection methods
3. The overall detection rate (26.7%) appears relatively low. Could this be due to the low abundance of ssDNA fragments, or is it related to variation in mutation-specific annealing temperatures? Further explanation would be helpful.
Response: The 26.7% detection rate reflects several biological and technical factors that are important to consider in the context of XPO1E571K mutation prevalence in lymphoma patients.
From a biological perspective, XPO1E571K mutations occur in specific lymphoma subtypes with varying frequencies - while this mutation is found in 28% of primary mediastinal B-cell lymphoma (PMBL) cases and 26% of classical Hodgkin lymphoma cases, it is rarely found in diffuse large B-cell lymphoma.
Additionally, cfDNA levels and mutation loads vary significantly between patients depending on disease stage, treatment status, tumor burden, and individual patient characteristics, with some patients having mutations below our detection threshold due to low circulating tumor DNA levels.
From a technical perspective, the variable quality and quantity of cfDNA in clinical samples can affect detection sensitivity, and individual sample variations in DNA integrity and concentration may influence amplification efficiency.
However, our detection rate is consistent with the expected prevalence of XPO1E571K mutations in lymphoma population, supporting the clinical relevance of our findings rather than indicating a technical limitation of the method.
4. For most experiments, the number of technical or biological replicates is not specified. Please include this information in the figure legends.
Response: Thank you for the suggestion. We have now clarified the number of technical replicates in the figure legends. Figures 5 and 6 were repeated in three independent experiments with consistent results (page 7, line 245; page 9, lines 276–277). Due to the limited quantity of patient-derived cfDNA samples, some other experiments were performed only once.